# TED—Trazodone Efficacy in Depression: A Naturalistic Study on the Efficacy of Trazodone in an Extended-Release Formulation Compared to SSRIs in Patients with a Depressive Episode—Preliminary Report

**DOI:** 10.3390/brainsci13010086

**Published:** 2023-01-02

**Authors:** Marcin Siwek, Aleksandra Gorostowicz, Adrian Andrzej Chrobak, Adrian Gerlich, Anna Julia Krupa, Andrzej Juryk, Dominika Dudek

**Affiliations:** 1Department of Affective Disorders, Jagiellonian University Medical College, Kopernika St. 21a, 31-501 Cracow, Poland; 2Department of Adult Psychiatry, Jagiellonian University Medical College, Kopernika St. 21a, 31-501 Cracow, Poland; 3Department of Adult, Child and Adolescent Psychiatry, University Hospital in Cracow, Kopernika St. 21a, 31-501 Cracow, Poland

**Keywords:** trazodone, SSRI, depression, insomnia, antidepressant drugs

## Abstract

These are the preliminary results of a 12-week non-randomized, open-label, non-inferiority study comparing the effectiveness of trazodone in an extended-release formulation (XR) versus SSRIs in the treatment of major depressive disorder (MDD). Participants (*n* = 76) were recruited, and 42 were assigned to the trazodone XR group and 34 to the SSRIs group. The choice of drug was based on clinical presentation and relied upon the attending physician. Assessments were made at five observation time points, at the following weeks: 0, and after 2, 4, 8, and 12 weeks. The evaluations included: symptoms of depression (MADRS, QIDS-clinician, and self-rated versions-primary study endpoints), anhedonia (SHAPS), anxiety (HAM-A), insomnia (AIS), psychosocial functioning (SDS), and therapeutic efficacy (CGI). At baseline, the trazodone group had significantly more severe depressive, anxiety, and insomnia symptoms and worse psychosocial functioning compared to the SSRIs group. After 12 weeks, trazodone XR was more effective than SSRIs in reducing the severity of insomnia and depression. There were no differences between the groups in the frequencies of therapeutic response and remission, which indicated the non-inferiority of the trazodone XR treatment. In conclusion, our results showed that in a “real world” setting, trazodone XR is effective in the treatment of patients with MDD.

## 1. Introduction

Depression affects 350 million people worldwide, and therefore, it has a considerable impact on the mortality and morbidity of the affected people, and it burdens society with high economic costs [1]. Antidepressants are one of the most commonly used medications—in 2015–2018, 13.2% of adult citizens in the USA reported having used an antidepressant medication in the last 30 days [2].

According to the recent network meta-analysis by Cipriani et al. (2018), the 21 studied antidepressants were more efficacious than placebos in the treatment of adult patients with major depressive disorder (MDD) [3]. Unfortunately, the effectiveness of pharmacotherapy in the treatment of depression is far from satisfying [4]. Selective serotonin reuptake inhibitors (SSRIs) are the most commonly prescribed antidepressants [5]. Despite their well-proven effectiveness, their therapeutic potential is limited. Not only are SSRIs partially effective in treating some symptomatic dimensions of depression, such as anhedonia, sleep disturbances, and sexual dysfunctions, but they may also exacerbate these symptoms [6,7,8,9]. Hence, SSRIs are not always an appropriate first-line MDD treatment. The majority of MDD treatment guidelines recommend antidepressants in monotherapy as the preferred therapeutic option and highlight the dangers of polypharmacy. However, the incomplete improvement of depressive symptoms achieved with SSRIs may force clinicians to prescribe adjunctive drugs in order to address the remaining symptoms.

Trazodone is an antidepressant from the 5-HT_2_ receptor antagonists and serotonin reuptake inhibitors (serotonin antagonists and reuptake inhibitor—SARI) class. Trazodone is approved for the treatment of major depressive episode in adults in many countries worldwide. In clinical practice, it is particularly recommended in MDD associated with insomnia and anxiety. Several studies have compared the effects of trazodone with other antidepressants, i.e., tricyclic drugs, SSRIs, and serotonin and norepinephrine reuptake inhibitors [10]. Their results have indicated that trazodone is an effective and well-tolerated antidepressant drug [11]. Compared to SSRIs, trazodone has been proven to improve sleep quality and anxiety sooner than other antidepressants [12]. Moreover, compared with fluoxetine and sertraline, trazodone has been associated with a significantly lower incidence of sexual dysfunction [13].

The new once-daily extended-release (XR) formulation of trazodone was developed to reduce both the peak plasma concentration and the dosing frequency in order to improve treatment adherence. The pharmacokinetic profile of trazodone XR is characterized by a slow increase in plasma concentration, with a single low- and delayed-peak, followed by a slow decrease. This is an advantage of the XR formula as high peak-plasma concentrations of trazodone may be associated with more side effects such as somnolence or low blood pressure, especially in the first weeks of treatment. These side effects can limit both the tolerance to trazodone and the compliance in patients with depression [11].

To date, numerous randomized case-control studies have been conducted comparing the efficacy of immediate-release or continued-release trazodone with other antidepressants. However, the number of articles assessing the trazodone XR formulation is limited to two studies [14,15]. While randomized controlled studies represent the “gold standard” of evaluating drug efficacy, it should be noted that the results of these studies are not always consistent with clinical practice as only approximately 20% of patients meet the inclusion criteria for these studies [16]. Therefore, naturalistic observations evaluating the effectiveness of drugs in a “real-world” setting are needed.

Our aim was to compare the effectiveness and tolerance of trazodone XR and SSRIs in the treatment of MDD. In this paper, we present the preliminary results of an ongoing naturalistic trial.

## 2. Materials and Methods

This was a 12-week, non-randomized, open-label, naturalistic trial that compared the effectiveness of trazodone XR and SSRIs in patients diagnosed with MDD and recurrent depressive disorder. Patients were assigned to one of the groups based on the analysis of the clinical presentation of depression, comorbidities, and potential drug interactions. The choice of study group (trazodone XR or SSRIs) or specific SSRI relied on the attending physician. The drug was selected based on a detailed analysis of the clinical manifestation of MDD and previous treatment history, following the guidelines of the Polish Psychiatric Association and the National Consultant for Adult Psychiatry in Poland [17].

Patients of the Department of Adult, Child, and Adolescent Psychiatry, University Hospital in Cracow were enrolled if they met the following inclusion criteria: age 18–65 years, diagnosis of a first episode of MDD according to the DSM-5 (Diagnostic and Statistical Manual of Mental Disorders 5th edition), or MDD in the course of recurrent depression.

The study’s exclusion criteria were: history or current episode of drug-resistant depression; a diagnosis of bipolar disorder, persistent mood disorder, organic mood disorder, or schizoaffective disorder; substance abuse (with the exception of nicotine and caffeine); pregnant or breastfeeding; non-consensual treatment; severe somatic diseases associated with renal, hepatic, circulatory, or respiratory failure; a diagnosis of the following selected neurological diseases: multiple sclerosis, neurodegenerative diseases, Parkinson’s disease, epilepsy, or dementia syndromes; and pharmacotherapy with drugs that induce the metabolism of antidepressants, e.g., rifampicin, glucocorticosteroids, phenytoin, and carbamazepine.

Patients were assigned to the one of the groups receiving (a) monotherapy with trazodone XR in a dose adjusted to the patient’s needs and clinical condition (dose of 150–300 mg/day) or (b) monotherapy with one of the following drugs from the group of SSRIs: sertraline (dose of 50–200 mg/day)), citalopram (dose of 20–40 mg/day), escitalopram (dose of 10–20 mg/day), and paroxetine (dose of 20–60 mg/day). The doses were adjusted to the patient’s needs and clinical condition.

The decision to the drop out of treatment was made by investigators and patients in the therapeutic alliance. Participants could decide to voluntarily discontinue the treatment at any point of the observation period, regardless of the reason. Investigators were able to stop further treatment in following situations: (1) if, from the perspective of both doctors and patients, there was no satisfactory or noticeable antidepressant effect at the subsequent observation points of treatment with an adequate dose (lack of effectiveness); (2) if the patient did not comply with the physician’s recommendations to such an extent that the proper assessment of the treatment effects was seriously biased (poor compliance, e.g., using subtherapeutic dose or irregular drug intake; and (3) when the severity of the side effects outweighed the benefits of the medication. Due to the naturalistic study design, we did not use specific criteria for dropping out based on cut-offs of the rating tools used in our project. Instead, this decision was made by the attending physician.

The study was approved by Bioethics Committee of the Jagiellonian University in Krakow (approval No. 1072.6120.113.2021). All participants provided a written informed consent.

### 2.1. Clinical Assessment

Assessments were carried out at 5 observation time points in the following weeks: 0 (at the beginning of the study) and after 2, 4, 8, and 12 weeks. During subsequent time points, the severity of depression, anhedonia, anxiety, and insomnia and the level of psychosocial functioning were evaluated. The following clinical tools were used:Montgomery–Åsberg Depression Rating Scale (MADRS); Quick Inventory of Depressive Symptomatology (QIDS)—clinician-rated (CR) and self-rated (SR) scales used to evaluate depressive symptomsHamilton Anxiety Rating Scale (HAM-A), used to measure anxietySnaith–Hamilton Pleasure Scale (SHAPS), used to assess the symptoms of anhedoniaSheehan Disability Scale (SDS), used to evaluate a patient’s psychosocial functioningAthens Insomnia Scale (AIS), used to measure the severity of insomniaClinical Global Impression Scale (CGI), used to assess the severity of symptoms and the treatment response

The primary endpoints were the changes in the severity of depression, as measured by the QIDS, QIDS-SR, and MADRS scales. The therapeutic response was defined as a decrease in the severity of depression by ≥50% on the QIDS-CR, QIDS-SR, or MADRS scales or a CGI-I score of 1 or 2 (“Very Much Improved” or “Much Improved”), as measured after 12 weeks of treatment with the selected drug. Remission was defined as achieving <6 points on the QIDS-CR or QIDS-SR scales or <10 points on the MADRS scale at week 12 of observation.

### 2.2. Statistical Analysis

Statistical analysis was performed on the data of the 76 participants recruited in this study. The baseline clinical and demographic variables were compared between the study groups (SSRIs vs. trazodone XR) using a 2-sample independent *t*-test or a Mann–Whitney U test for continuous variables and an χ^2^ test for the categorical variables. The distribution of the continuous variables was examined with the Shapiro–Wilk test. The continuous variables are presented as means and standard deviations or medians and percentiles (25th and 75th) and the categorical variables are presented as frequencies.

Changes in the QIDS-CR, QIDS-SR, and MADRS total scores from baseline across subsequent timepoints were the primary study endpoints. The Linear Mixed-Effects Model (MMRM—a mixed model for repeated measures) was used for the statistical analysis via the lmer function from the lme4 package in R (version R 4.2.1 [18]). The model included the time points of measurement (0, 2, 4, 8, and 12 weeks) and the treatment groups (SSRIs or trazodone XR) as the fixed effects and the participants as the random effects (with Restricted Maximum Likelihood (REML) applied). The effects of time, treatment, and time multiplied by treatment (interaction) on the dependent variables (QIDS-CR, QIDS-SR, and MADRS scores) were assessed. The effect size was calculated as a partial eta squared for the interaction. Between-group comparisons (SSRIs vs. trazodone XR) were calculated for the estimated marginal means at each time point.

The evaluation of the secondary outcomes (scores in the SHAPS, HAM-A, AIS, and SDS scales) was performed analogously to the method described above.

Additional analysis was performed with the same method for all the outcomes with the duration of the previous psychiatric treatment included as a covariate in the model.

For both primary and secondary outcomes and mean body weights, changes across time were evaluated for each treatment group separately by repeated measures ANOVA. The mean changes in outcomes scores and weight after 12 weeks of treatment (compared to baseline) were calculated.

Reliability analysis was performed for the items in the QIDS-CR, QIDS-SR, MADRS, HAMA, SDS, SHAPS, and AIS (measured at baseline) scales. Internal consistency reliability was conducted using Cronbach’s α. An α above 0.7 was considered acceptable.

The relations between the treatment group and the 1) therapeutic response (defined as a ≥50% reduction in MADRS, QIDS-CR, or QIDS-SR scores after 12 weeks of treatment) and 2) remission (defined as <6 points in the QIDS-CR or QIDS-SR score and <10 points in the MADRS score at 12 weeks) were examined by the χ^2^ tests. The relations between the treatment group and 1) achievement of a CGI-I score of 1 or 2 (much or very much improved) after 12 weeks of therapy were evaluated with the use of a Fisher’s exact test.

For all tests, the statistical significance was defined as a *p*-value of < 0.05. Because of the preliminary nature of this report (the study is ongoing and a larger sample size is being recruited), which was intended to perform an exploratory statistical analysis, no correction for multiple testing was applied.

## 3. Results

Out of the 83 initially enrolled subjects, 7 patients did not consent to participate in the study. The study included 76 patients aged 18–65, of both sexes, of Caucasian origin, diagnosed with MDD according to the DSM-5 classification.

Patients were assigned to one of two groups, the first, treated with trazodone XR, or the second, treated with an SSRI (Figure 1). Of the total subjects, 42 were started on trazodone XR monotherapy and 34 on SSRIs. In the SSRIs group, 25 subjects received sertraline and 9 received escitalopram. Nine patients assigned to the SSRI group (26.5%) and eleven assigned to the trazodone XR group (26.2%) dropped out before completion of the 12-week trial. The main reasons for treatment discontinuation were lack of effectiveness and poor compliance to the study procedures.

The mean dose of the SSRIs used (defined as fluoxetine equivalent in mg) was 21.7 mg, and the mean dose of trazodone XR was 209.4 mg. The mean age of the subjects using trazodone XR was 35.4 ± 12.6 years, and for those treated with SSRIs, it was 39.1 ± 12.6 years.

A comparison of the baseline group characteristics is presented in Table 1. There were no significant differences between the treatment groups in age, sex, weight, alcohol consumption, smoking, presence of somatic comorbidities, or psychotherapy use. Patients in the trazodone group had a significantly longer history of psychiatric treatment and higher scores in the QIDS-CR, QIDS-SR, HAMA, AIS, and SDS scales. There were no statistically significant differences between studied groups in the baseline scores of the MADRS and SHAPS scales.

Table 2 presents the results of the MMRM models for each outcome measure. There was a statistically significant effect of interaction between the time and treatment type for scores in the QIDS-CR (F(4, 242.1) = 3.65, *p* = 0.007), QIDS-SR (F(4, 213.9) = 3.7, *p* = 0.006), HAM-A (F(4, 236.31) = 4, *p* = 0.004), AIS (F(4, 244.9) = 2.6, *p* < 0.001), and SDS (F(4, 209.4) = 2.47, *p* = 0.046) scales. The effect sizes for the time–treatment interaction (measured by the partial eta squared η2) were large for the AIS (η2 = 0.17) scale, moderate for the QIDS-CR, QIDS-SR, and HAMA (η2 = 0.06 for all) scales, and small for the SDS (η2 = 0.05) scale.

In Table 3, the estimated marginal means for each outcome measure are presented separately at each time point (baseline and 2, 4, 8, and 12 weeks) with appropriate *p*-values for comparisons between the SSRIs and trazodone groups. Statistically significant differences were observed for the QIDS-CR scores at 12 weeks (SSRI 5.2 vs. trazodone 2.7, *p* = 0.048); QIDS-SR scores at baseline (SSRI 13 vs. trazodone 15.6, *p* = 0.044); HAM-A scores at baseline (SSRI 18.9 vs. trazodone 22.8, *p* = 0.017); AIS scores at baseline (SSRI 9.4 vs. trazodone 14.5, *p* < 0.001), at 4 weeks (SSRI 8.3 vs. trazodone 5.6, *p* = 0.025), and at 12 weeks (SSRI 6.4 vs. trazodone 3, *p* = 0.017); and SDS scores at baseline (SSRI 15.5 vs. trazodone 20.3, *p* = 0.014) and at 2 weeks (SSRI 13.3 vs. trazodone 18.2, *p* = 0.014).

Line graphs for each outcome measured with mean values represented on the *y*-axis and sequential time points on the *x*-axis, with separate lines for each treatment option (SSRIs or trazodone), are available in Appendix A.

Table 4 presents the results of the MMRM models for each outcome measured, with the duration of previous psychiatric treatment as a covariate. There was a statistically significant effect of the interaction between time and treatment type for the scores in the QIDS-CR (F(4, 196.8) = 2.47, *p* = 0.046) and AIS (F(4, 197.4) = 7.18, *p* < 0.001) scales. The effect sizes for the time–treatment interaction (measured by the partial eta squared η2) were moderate for the AIS (η2 = 0.13) scale and small for the QIDS-CR (η2 = 0.05) scale.

Table 5 shows the results of the repeated measures ANOVA for each outcome measured across time separately for SSRIs and trazodone XR groups. Significant changes from baseline were observed for all primary and secondary outcomes across both treatment groups. No statistically meaningful changes in weight were observed across time.

Reliability was acceptable for all scales, with the Cronbach’s α values for the QIDS-CR, QIDS-SR, MADRS, HAMA, SDS, SHAPS, and AIS scales equal to 0.747, 0.780, 0.771, 0.764, 0.925, 0.913, and 0.916, respectively.

Table 6 shows that no statistically significant differences were observed between the SSRIs and trazodone XR groups in the prevalence of therapeutic response or remission (both measured by MADRS, QIDS-CR, and QIDS-SR scores at 12 weeks) and global improvement (assessed at 12 weeks by the CGI-I scale).

## 4. Discussion

In the analysed groups, both trazodone XR and the SSRIs (sertraline and escitalopram) were effective in the treatment of the symptoms of depression, anxiety, anhedonia, and sleep disorders. Improvements in functioning were also observed for both groups (as measured by the SDS scale). Neither of the treatment options induced any clinically significant changes in body weight. Discontinuation rates before completion of the 12-week study period were comparable at 26.5% for the SSRIs group and 26.2% for the trazodone XR group. Both therapies were well-tolerated as only one patient in each group discontinued the treatment due to adverse effects (a patient on trazodone discontinued treatment due to excessive sleepiness and a patient on sertraline discontinued treatment because of hypomania).

At baseline, the patients receiving trazodone had significantly more severe depressive and anxiety symptoms, as well as more pronounced sleep disturbances, and worse psychosocial functioning compared to the SSRIs group. Despite this, the use of trazodone XR was more effective than SSRIs in reducing depressive symptoms (measured by QIDS-CR scores) and reducing the severity of insomnia (measured by AIS scores), even when controlling for the duration of previous psychiatric treatment as a covariate. In the analysis of the results measured with the MADRS and SHAPS scales, no statistically significant differences were observed between the drug groups; however, the patients treated with trazodone XR achieved a greater numerical improvement than those treated with SSRIs (decrease from baseline to 12 weeks in MADRS and SHAPS total scores of 24.8 and 6.2 for trazodone XR, respectively, vs. 19.5 and 3.4 for SSRIs, respectively). It is worth noticing that we observed statistically significant differences between the treatment arms when measuring depressive symptoms with the QIDS-CR scale, but not with the MADRS scale. Depression is a highly heterogenous condition. The analysis of the most commonly used depression rating tools has shown that the mean overlap among the scales is low [19]. Using only one rating scale can limit the replicability of the results, and thus, it is important to assess the outcome with different tools [19].

The patients treated with trazodone XR achieved the therapeutic response as measured by the QIDS-CR scale (approximately 84% vs. 64%, respectively) and QIDS-SR scale (70% vs. 65%, respectively) more often than those treated with SSRIs, with similar outcomes for symptomatic remission as measured by the QIDS scale (approximately 84% vs. 64%, respectively), the MADRS scale (approximately 81% vs. approximately 71%), and the QIDS-SR scale (69% vs. 60%). However, these differences did not reach statistical significance, which may have been due to the small sample size. Both the trazodone XR and SSRIs groups showed high percentages of therapeutic response and remission when compared with previous studies. In one of the biggest “real-world” studies in which the effectiveness of citalopram in patients with depression was assessed (the STAR*D trial), the response rate after 14 weeks of observation was only 47%, while remission rates were 28% and 33% (as measured by the HAM-D and QIDS-SR scales, respectively) [20]. Clinical trials in patients with MDD have shown that trazodone has comparable efficacy to other antidepressants [21]. Trazodone was observed to be equal in efficacy when compared with older antidepressants (e.g., amitriptyline [22,23], imipramine [24], clomipramine [15], and dothiepin [25]). Similar improvements in depressive symptoms were shown when trazodone was compared to fluoxetine [26], sertraline [27], paroxetine [28], venlafaxine [29], and bupropion [30]. One study reported a significant difference in antidepressant efficacy in favour of extended-release venlafaxine (compared to trazodone) after 8 weeks of treatment [14]. Another study compared trazodone with mirtazapine, indicating that treatment with the latter showed a significantly greater improvement in depressive symptoms [31]. Only two of the above-mentioned trials compared the efficacy of the trazodone XR formulation with other antidepressants [14,15]. Almost all of these studies were double-blind, randomized trials in which outcomes were measured by the Hamilton Depression Rating Scale, MADRS scale, or CGI scale. To the best of our knowledge, no previous studies have compared trazodone with SSRIs using the QIDS/QIDS-SR, HAM-A, SDS, or SHAPS scales. Therefore, our study provides a significant advance in the knowledge on the efficacy of trazodone XR in MDD due to the precise assessment of the (1) changes in different depression scales, (2) influence on the hedonic tone, and (3) improvement of functioning. This if of crucial importance to “real-life” settings where the goal of treatment is not symptomatic remission, but rather, functional remission, which is better approximated by the assessment of positive affect or a direct measurement level of functioning than the depression scales, which focus more on the remission of negative affect (i.e., the HAM-D [32]).

As already mentioned in the introduction, weight gain and sleep problems are one of the most common reasons for treatment non-adherence in patients treated with antidepressants. Thus, the high effectiveness of treating insomnia and the lack of significant impact on body weight make trazodone XR a valuable treatment option.

We are aware of several limitations in our study: (a) the small number of participants; (b) the open-label, non-randomized design—a lack of randomization can be the reason for differences in the baseline symptoms severity and clinical characteristics between the treatment groups and their non-equal distribution; (c) the single-centre study, (d) the doses of antidepressants, which varied between patients; (e) the different medications used in the SSRIs group; however, the study is ongoing and the presented data are preliminary; and (f) the naturalistic study design, where we did not use specific criteria for dropping out based on cut-offs of the rating tools used in our project; instead, this decision was made by the attending physician. More participants are currently being recruited, and therefore, the analysis of the final data will provide more reliable results. Further, the design of the study made it easier to perform in an actual clinical setting, with “real-world” patients included.

## 5. Conclusions

In conclusion, we have shown that trazodone XR in monotherapy is an effective and well-tolerated medication for patients with MDD. Compared to SSRIs, treatment with trazodone XR was associated with a greater reduction in insomnia and with a larger improvement in depressive symptoms. This work shows that trazodone XR is not inferior to SSRIs in the treatment of MDD. Further studies using double-blinded, randomized design and varied outcome measures are needed.

## Figures and Tables

**Figure 1 brainsci-13-00086-f001:**
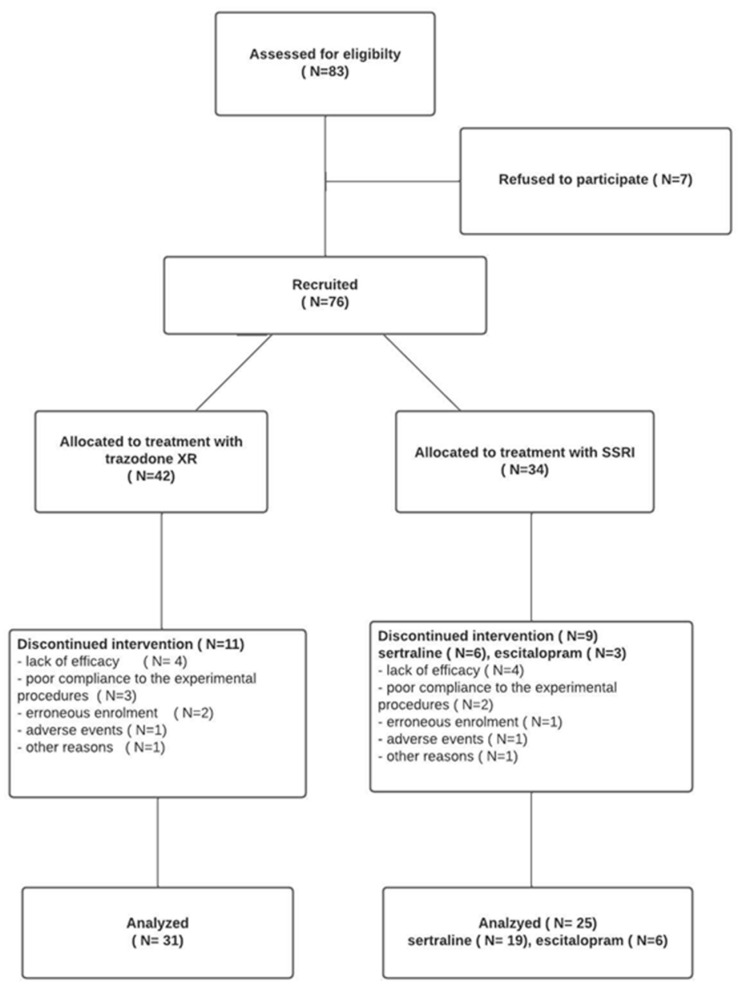
Flowchart of the study.

**Table 1 brainsci-13-00086-t001:** Baseline group characteristics. SD—standard deviation; MADRS—Montgomery–Åsberg Depression Rating Scale; HAM-A—Hamilton Anxiety Rating Scale; QIDS-CR—Quick Inventory of Depressive Symptomatology, clinician rating; QIDS-SR—Quick Inventory of Depressive Symptomatology, self rating; SHAPS—Snaith–Hamilton Pleasure Scale; AIS—Athens Insomnia Scale; CGI—Clinical Global Impression Scale; XR—extended-release formulation.

	SSRI (*n* = 34)	Trazodone XR (*n* = 42)	*p*
Sex (% female)	64.7	61.9	0.801 ^a^
Age (in years): mean (SD)	39.1 (12.6)	35.4 (12.8)	0.230 ^b^
Weight (in kilograms): median (25th; 75th percentile)	70 (58.5; 76)	68 (60; 80)	0.945 ^c^
Duration of previous psychiatric treatment (in months): median (25th; 75th percentile)	0 (0; 9.75)	12 (0; 60)	0.044 ^c^
Alcohol consumption (% yes)	69.2	76.9	0.489 ^a^
Smoking (% yes)	12	23.1	0.338 ^a^
Somatic comorbidities (% yes)	43.3	40	0.779 ^a^
Psychotherapy (% yes)	44	52.5	0.505 ^a^
QIDS-CR: mean (SD)	12.7 (4.1)	14.8 (4.3)	0.033 ^b^
QIDS-SR: mean (SD)	13.3 (4.8)	15.9 (4.6)	0.049 ^b^
MADRS: mean (SD)	27 (7.8)	28.7 (7)	0.345 ^b^
SHAPS: median (25th; 75th percentile)	6 (2; 10.3)	8 (3.3; 12)	0.433 ^c^
HAMA: mean (SD)	18.9 (7.3)	22.9 (7.6)	0.027 ^b^
AIS: mean (SD)	9.4 (5.4)	14.5 (6.1)	<0.001 ^b^
SDS: mean (SD)	15.3 (7.8)	19.8 (8.1)	0.031 ^b^

^a^, chi-square test; ^b^, independent sample *t*-test; ^c^, Mann–Whitney U test.

**Table 2 brainsci-13-00086-t002:** Results of mixed-effect model showing the significance levels and effect sizes (partial eta squared) for all outcomes. MADRS—Montgomery–Åsberg Depression Rating Scale; HAM-A—Hamilton Anxiety Rating Scale; QIDS-CR—Quick Inventory of Depressive Symptomatology, clinician rating; QIDS-SR—Quick Inventory of Depressive Symptomatology, self rating; SHAPS—Snaith-Hamilton Pleasure Scale; AIS—Athens Insomnia Scale; and CGI—Clinical Global Impression Scale.

	Time Effect, *p*	Treatment Effect, *p*	Time–Treatment Effect, *p*	Partial Eta Squared for Interaction (with 95% CI)
MDRS	<0.001	0.9004	0.1724	0.03 (0.00–0.06)
QIDS-CR	<0.001	0.782	0.007	0.06 (0.01–0.11)
QIDS-SR	<0.001	0.820	0.006	0.06 (0.01–0.12)
HAM-A	<0.001	0.546	0.004	0.06 (0.01–0.12)
AIS	<0.001	0.640	<0.001	0.17 (0.09–0.25)
SDS	<0.001	0.123	0.046	0.05 (0.00–0.10)
SHAPS	<0.001	0.643	0.145	0.03 (0.00–0.06)

**Table 3 brainsci-13-00086-t003:** Between-group comparisons for each time point. Values are presented as estimated marginal means with 95% confidence intervals. Emmean—estimated marginal mean; T-XR—trazodone extended-release formulation; MADRS—Montgomery–Åsberg Depression Rating Scale; HAM-A—Hamilton Anxiety Rating Scale; QIDS-CR—Quick Inventory of Depressive Symptomatology, clinician rating; QIDS-SR—Quick Inventory of Depressive Symptomatology, self rating; SHAPS—Snaith–Hamilton Pleasure Scale; AIS—Athens Insomnia Scale; and CGI—Clinical Global Impression Scale.

	Baseline Emmean (95% CI)			2 Weeks Emmean (95% CI)			4 Weeks Emmean (95% CI)			8 Weeks Emmean (95% CI)			12 Weeks Emmean (95% CI)		
	SSRI	T-XR	*p*	SSRI	T-XR	*p*	SSRI	T-XR	*p*	SSRI	T-XR	*p*	SSRI	T-XR	*p*
MDRS	27.1 (24.2–29.9)	28.5 (24.2–29.9)	0.456	20.7 (17.6–23.7)	22.6 (19.9–25.4)	0.356	13.6 (10.5–16.7)	14.6 (11.9–17.3)	0.629	10.4 (7.1–13.7)	8.3 (5.5–11.2)	0.345	8.2 (5–11.5)	5 (2.1–7.9)	0.152
QIDS-CR	12.7 (11.1–14.3)	14.9 (13.4–16.3)	0.057	10 (8.3–11.8)	11.1 (9.6–12.7)	0.348	7.5 (5.7–9.4)	7.3 (5.7–8.9)	0.826	6 (4.2–7.9)	4.4 (2.8–6.1)	0.206	5.2 (3.4–7.1)	2.7 (1–4.4)	0.048
QIDS- SR	13 (11.2–14.8)	15.6 (13.8–17.4)	0.044	10.6 (8.8–12.4)	12.8 (11.2–14.4)	0.082	8.7 (6.9–10.5)	8.2 (6.5–9.8)	0.682	6.7 (4.8–8.6)	5.5 (3.8–7.3)	0.364	6 (4.1–7.9)	3.9 (2.1–5.6)	0.105
HAM-A	18.9 (16.6–21.2)	22.8 (20.6–25)	0.017	12.1 (9.7–14.6)	14.9 (12.72–17.1)	0.099	7.9 (5.4–10.4)	8.4 (6.2–10.6)	0.764	6.2 (3.7–8.7)	4.6 (2.3–6.8)	0.343	4.9 (2.4–7.5)	3.2 (0.9–5.5)	0.318
AIS	9.4 (7.8–11.1)	14.5 (13–16)	<0.001	9 (7.3–10.8)	9.5 (8–11.1)	0.669	8.3 (6.6–10.1)	5.6 (4–7.2)	0.025	6.1 (4.3–7.9)	4.3 (2.6–6)	0.152	6 (4.2–7.9)	3 (1.3–4.7)	0.017
SDS	15.5 (12.7–18.4)	20.3 (17.8–22.8)	0.014	13.3 (10.3–16.3)	18.2 (15.7–20.7)	0.014	10.8 (7.7–13.8)	11.6 (9–14.1)	0.695	7.9 (4.9–11)	9.2 (6.6–11.9)	0.527	6.5 (3.4–9.6)	6.2 (3.5–8.8)	0.877
SHAPS	6.5 (4.9–8)	7.4 (6–8.8)	0.400	5.4 (3.8–7.1)	6.5 (5.1–8)	0.315	5.3 (3.7–7)	3.9 (2.4–5.5)	0.220	3.6 (1.9–5.3)	2.8 (1.2–4.3)	0.481	3.5 (1.7–5.2)	2 (0.4–3.6)	0.210

**Table 4 brainsci-13-00086-t004:** Results of the mixed-effect model, with the duration of previous psychiatric treatment as a covariate, showing the significance levels and effect sizes (partial eta squared) for all outcomes. MADRS—Montgomery–Åsberg Depression Rating Scale; HAM-A—Hamilton Anxiety Rating Scale; QIDS-CR—Quick Inventory of Depressive Symptomatology, clinician rating; QIDS-SR—Quick Inventory of Depressive Symptomatology, self rating; SHAPS—Snaith–Hamilton Pleasure Scale; and AIS—Athens Insomnia Scale.

	Time Effect, *p*	Treatment Effect, *p*	Time–Treatment Effect, *p*	Partial Eta Squared for Interaction (with 95% CI)
MDRS	0.022	0.851	0.368	0.02 (0.00–0.06)
QIDS-CR	<0.001	0.931	0.046	0.05 (0.00–0.10)
QIDS-SR	<0.001	0.534	0.065	0.05 (0.00–0.11)
HAM-A	<0.001	0.461	0.057	0.05 (0.00–0.10)
AIS	<0.001	0.458	<0.001	0.13 (0.04–0.21)
SDS	<0.001	0.251	0.151	0.04 (0.00–0.09)
SHAPS	<0.001	0.829	0.148	0.03 (0.00–0.06)

**Table 5 brainsci-13-00086-t005:** Mean changes in outcomes scores and weight after 12 weeks of treatment (compared to baseline), as measured by repeated measures ANOVA. MADRS—Montgomery–Åsberg Depression Rating Scale; HAM-A—Hamilton Anxiety Rating Scale; QIDS-CR—Quick Inventory of Depressive Symptomatology, clinician rating; QIDS-SR—Quick Inventory of Depressive Symptomatology, self rating; SHAPS—Snaith–Hamilton Pleasure Scale; AIS—Athens Insomnia Scale; and CGI—Clinical Global Impression Scale.

	SSRI (Mean Change: After 12 Weeks–Baseline Score; 95% CI)	*p*	Trazodone XR (Mean Change: After 12 Weeks–Baseline Score; 95% CI)	*p*
MADRS	−19.5 (−23.9; −15)	<0.001	−24.8 (−29.1; −20.5)	<0.001
QIDS-CR	−7 (−9.62; −4.38)	<0.001	−12.8 (−15.5; −10.1)	<0.001
QIDS-SR	−6.1 (−9.3; −3)	<0.001	−12.4 (−16.2; −8.6)	<0.001
HAM-A	−13.5 (−17; −10.1)	<0.001	−20.6 (−23.3; −18)	<0.001
AIS	−3.7 (−5.7; −1.8)	<0.001	−12.5 (−15.4; −9.5)	<0.001
SDS	−9.4 (−14.2; −4.5)	<0.001	−16 (−20.9; −11.1)	<0.001
SHAPS	−3.4 (−5.7; −1.2)	0.027	−6.2 (−8.7; −3.8)	<0.001
Weight (in kg)	0.1 (−1; 1.1)	0.746	−0.15 (−1.8; 1.5)	0.629

**Table 6 brainsci-13-00086-t006:** Comparison of the frequencies of therapeutic response, remission (measured by QIDS, QIDS-SR, and MADRS scores) and clinical improvement (measured by CGI-I score) between the patients treated with SSRIs and those treated with trazodone XR after 12 weeks of treatment. MADRS—Montgomery–Åsberg Depression Rating Scale; QIDS-CR—Quick Inventory of Depressive Symptomatology, clinician rating; QIDS-SR—Quick Inventory of Depressive Symptomatology, self rating; and CGI—Clinical Global Impression Scale.

	SSRI	Trazodone XR	*p*
Therapeutic response (≥50% reduction in QIDS score after 12 weeks), % of patients	64	83.9	0.088 ^a^
Therapeutic response (≥50% reduction in MADRS score after 12 weeks), % of patients	79.2	79.3	0.999 ^b^
Therapeutic response (≥50% reduction in QIDS-SR score after 12 weeks), % of patients	65	70	0.739 ^a^
Remission (<6 points on QIDS-CR scale) after 12 weeks, % of patients	64	83.9	0.088 ^a^
Remission (<10 points on MADRS scale) after 12 weeks, % of patients	70.8	80.6	0.396 ^a^
Remission (<6 points on QIDS-SR scale) after 12 weeks, % of patients	60	69	0.492 ^a^
CGI-I score of 1 or 2 after 12 weeks of treatment, % of patients	77.3	77.3	>0.999 ^a^

^a^, chi-square test; ^b^, Fisher’s exact test.

## Data Availability

The study data is available on request.

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
