# Peer review of "TED—Trazodone Efficacy in Depression: A Naturalistic Study on the Efficacy of Trazodone in an Extended-Release Formulation Compared to SSRIs in Patients with a Depressive Episode—Preliminary Report"

_brainsci, 2023, doi:10.3390/brainsci13010086_

Round 1

Reviewer 1 Report

The manuscript reports an interesting study about the effects of a specific drug formulation in a real-world setting. The paper is well-written and presents data that are interesting for the readers. However, I have some concerns about the methodology that authors might use to improve their paper:

- please include the reliability tests for your questionnaires.

- it is unclear to me how the SSRI was decided for the control group. Is it not possible that the authors decided on the wrong drug for control? I think this is a relevant point that might compromise all their conclusions. 

- The clinical characteristics of the samples are not equally distributed. The trazodone group presented a longer duration of treatment. This is a not-randomized. This is another critical point because it might have a role in the results. At least the authors should provide an ANCOVA analysis with this data as a covariate.

Author Response

Dear Reviewer,

Thank you for your suggestions. Attached you will find our responses and the corrected manuscript.

Yours respectfully,

Authors

Reviewer 1

Suggestion 1:

“- please include the reliability tests for your questionnaires.”

Answer 1:

Reliability analysis was performed, as suggested.

Following text was added:

  1. section 2.2 Statistical analysis, lines 171-173:

“Reliability analysis was performed for the items in QIDS-CR, QIDS-SR, MADRS, HAMA, SDS, SHAPS, AIS (measured at baseline). Internal consistency reliability was conducted using Cronbach’s α. An α above 0.7 was considered acceptable. “

  1. section Results, lines 277-279: “Reliability was acceptable for all scales, with Cronbach’s α for QIDS-CR, QIDS-SR, MADRS, HAMA, SDS, SHAPS and AIS equal to 0.747, 0.780, 0.771, 0.764, 0.925, 0.913 and 0.916 respectively.”

Suggestion 2

“- it is unclear to me how the SSRI was decided for the control group. Is it not possible that the authors decided on the wrong drug for control? I think this is a relevant point that might compromise all their conclusions. “

Answer 2:

Following fragment has been added (page 2, Materials and Methods, lines 88-90):

“The drug was selected based on a detailed analysis of the clinical manifestation of MDD and previous treatment history, following the guidelines of the Polish Psychiatric Association and the National Consultant for Adult Psychiatry in Poland [17].”

Suggestion 2:

“- The clinical characteristics of the samples are not equally distributed. The trazodone group presented a longer duration of treatment. This is a not-randomized. This is another critical point because it might have a role in the results. At least the authors should provide an ANCOVA analysis with this data as a covariate.”

Answer 2:

Following fragment has been edited (page 14, paragraph 1, lines 332-335):

“We are aware of several limitations of our study: a) small number of participants; b) open-label, non-randomized design – lack of randomization can be the reason of differences in baseline symptoms severity and clinical characteristics between treatment groups and their non-equal distribution”

We have built an additional model with the duration of previous psychiatric treatment included as a covariate, in accordance with the suggestion.  In section “Statistical analysis”, lines 154-155 we have added a following fragment: “Additional analysis was performed with the same method for all the outcomes with duration of the previous psychiatric treatment included as a covariate in the model.”

In the Results section (lines 232-237) the following text has been added: “Table 4 presents results of the MMRM models for each outcome measure with the duration of previous psychiatric treatment as a covariate. There was a statistically significant effect of interaction between time and treatment type for scores in: QIDS-CR [F(4, 196.8)= 2.47, p=0.046] and AIS [F(4, 197.4)= 7.18, p<0.001]. Effect sizes for time x treatment interaction (measured by the partial-eta squared η2) were moderate for AIS (η2 = 0.13) and small for QIDS-CR (η2 = 0.05).” 

Additional table (Table 4) with results of the new model has been included in the Results section.

In the Discussion part we added the underlined part of the sentence (lines 285-286): “In spite of this, the use of trazodone XR was more effective than SSRIs in reducing depressive symptoms (measured by QIDS-CR) and the severity of insomnia (measured by AIS), even when controlling for the duration of previous psychiatric treatment as a covariate.”

Additional changes:

Article underwent extensive English revision.

Reviewer 2 Report

This is a report of a still ongoing study (as preliminary findings) enrolling patients with MDD treated with Trazodone XR or SSRI's. By stating that, I have to emphasize that treatment with Trazodone is nothing new, and is a validated MDD treatment option. Sure, the authors included XR formulation that has not received that much attention manuscript-wise. Still, it is nothing new. Also, they compared Trazodone XR with SSRI's (four of them), while the mechanism of action is shared (blockade od SERT), along with 5HT2 effects of Trazodone.

One of the biggest problems I have with this study is the sample size. I am aware it is an ongoing study, but salami style to publication is not my cup of tea. It could be all right for a conference paper.

The text has some flaws (e.g. "In the SSRIs group 25 subjects received sertraline and 91 escitalopram", while the results are presented too exhaustively, especially diagrams which should be text described and possibly used as supp. material. Also, from table 1 I can gather that patients receiving Trazdone-XR had 12 months of treatment at inclusion, while those treated with SSRI's had 0. That hardly feels like the groups were balanced.

In the end, I feel after reading this article that I learned nothing new, while the size of just over 70 patients  (split into two groups) for MDD which has well-known high incidence/prevalence is significantly underpowered to draw firm conclusions.

Author Response

Dear Reviewer,

Thank you for your suggestions. Attached you will find our responses and the corrected manuscript.

Yours respectfully,

Authors

Reviewer 2

Suggestion 1:

This is a report of a still ongoing study (as preliminary findings) enrolling patients with MDD treated with Trazodone XR or SSRI's. By stating that, I have to emphasize that treatment with Trazodone is nothing new, and is a validated MDD treatment option. Sure, the authors included XR formulation that has not received that much attention manuscript-wise. Still, it is nothing new. Also, they compared Trazodone XR with SSRI's (four of them), while the mechanism of action is shared (blockade od SERT), along with 5HT2 effects of Trazodone.

Answer 1:

To the best of our knowledge, no previous studies compared trazodone XR with SSRI using QIDS/QIDS-SR, HAM-A, SDS or SHAPS scales. Therefore, our study provides a significant advance in the knowledge on efficacy of trazodone XR in MDD due to the precise assessment of 1) the changes in different depression scales, 2) influence on the hedonic tone, 3) improvement of functioning.

Those points have been mentioned in discussion section (page 10, lines x).

Suggestion 2:

“One of the biggest problems I have with this study is the sample size. I am aware it is an ongoing study, but salami style to publication is not my cup of tea. It could be all right for a conference paper.”

Answer 2:

Abovementioned limitations have been listed in following paragraph (lines 332-340):

“We are aware of several limitations of our study: a) small number of participants; b) open-label, non-randomized design – lack of randomization can be the reason of differences in baseline symptoms severity and clinical characteristics between treatment groups and their non-equal distribution; c) single centre study, d) doses of antidepressants varied between patients; e) different medications were used in SSRIs group. However, the study is ongoing and the presented data are preliminary. More participants are being recruited and therefore the analysis of final data will provide more re-liable results. Besides, the design of the study made it easier to perform in actual clinical setting, with “real-world” patients included”

Suggestion 3:

“The text has some flaws (e.g. "In the SSRIs group 25 subjects received sertraline and 91 escitalopram",

Answer 3:

Abovementioned fragment has been corrected (page 4, line 179):

“In the SSRIs group 25 subjects received sertraline and 9 escitalopram.”

Suggestion 4

While the results are presented too exhaustively, especially diagrams which should be text described and possibly used as supp. material.

Answer 4:

Diagrams have been moved to supp. materials (Fig. S1).

Following fragment has been modified (page 8, lines 228-230):

“Line graphs for each outcome measure with mean values represented on Y-axis and sequential time points on X-axis, with separate lines for each treatment option (SSRIs or trazodone) are available in supplementary materials (Fig. S1”

Information about supplementary materials has been added: (page 11)

“Supplementary Materials: The following supporting information can be downloaded at: www.mdpi.com/xxx/s1, Figure S1: Line graphs for each outcome measure evaluated in the study.”

Suggestion 5:

Also, from table 1 I can gather that patients receiving Trazdone-XR had 12 months of treatment at inclusion, while those treated with SSRI's had 0. That hardly feels like the groups were balanced.

Answer 5:

Following fragment has been edited (page 11, lines 332-335):

“We are aware of several limitations of our study: a) small number of participants; b) open-label, non-randomized design – lack of randomization can be the reason of differences in baseline symptoms severity and clinical characteristics between treatment groups and their non-equal distribution”

We have built an additional model with the duration of previous psychiatric treatment included as a covariate.  In section “Statistical analysis”, lines 154-155 we have added the following fragment: “Additional analysis was performed with the same method for all the outcomes with duration of the previous psychiatric treatment included as a covariate in the model.”

In the Results section (lines 232-237) the following text has been added: “Table 4 presents results of the MMRM models for each outcome measure with the duration of previous psychiatric treatment as a covariate. There was a statistically significant effect of interaction between time and treatment type for scores in: QIDS-CR [F(4, 196.8)= 2.47, p=0.046] and AIS [F(4, 197.4)= 7.18, p<0.001]. Effect sizes for time x treatment interaction (measured by the partial-eta squared η2) were moderate for AIS (η2 = 0.13) and small for QIDS-CR (η2 = 0.05).” 

Additional table (Table 4) with results of the new model has been included in the Results section.

In the “Discussion” part we added the underlined part of the sentence (lines 285-286): “In spite of this, the use of trazodone XR was more effective than SSRIs in reducing depressive symptoms (measured by QIDS-CR) and the severity of insomnia (measured by AIS), even when controlling for the duration of previous psychiatric treatment as a covariate.”

Suggestion 6:

“In the end, I feel after reading this article that I learned nothing new, while the size of just over 70 patients (split into two groups) for MDD which has well-known high incidence/prevalence is significantly underpowered to draw firm conclusions.”

Answer 6:

“We have responded to the abovementioned points in answer nr 1 and 2.”

Additional changes:

Article underwent extensive English revision.

Reviewer 3 Report

This thesis demonstrates that trazodone XR is effective in the treatment of patients with MDD. The statistical experimental design of the paper is reasonable and the statistical methods used are correct.

Author Response

Dear Reviewer,

Thank you for your suggestions. Attached you will find our responses and the corrected manuscript.

Yours respectfully,

Authors

Reviewer 3

“This thesis demonstrates that trazodone XR is effective in the treatment of patients with MDD. The statistical experimental design of the paper is reasonable and the statistical methods used are correct.”

Answer:

We would like to thank you for the positive feedback about our study.

Additional changes:

Article underwent extensive English revision.

Round 2

Reviewer 1 Report

I think the authors have improved their manuscript and have replied to my concerns. I think the data might have an impact in clinical practice, so I suggest to publish the paper.

Author Response

We would like to thank you for the positive feedback about our study.

Reviewer 2 Report

-

Author Response

We would like to thank you for the review of our study.